



# Physical drivers of the nitrate seasonal variability in the Atlantic cold tongue

Marie-Hélène Radenac[1], Julien Jouanno[1], Christine Carine Tchamabi[1][*], Mesmin Awo[1,2,3], Bernard Bourlès[4], Sabine Arnault[5], Olivier Aumont[5]

[1]LEGOS, IRD-Université Paul Sabatier-Observatoire Midi-Pyrénées, Toulouse, 31400, France
[2]Nansen-Tutu Centre for Marine Environmental Research, Department of Oceanography, University of Cape Town, Cape Town, South Africa
[3]LHMC, IRHOB, IRD, Cotonou, Bénin
[4]IRD, US191 "Instrumentation, Moyens Analytiques, Observatoires en Géophysique et Océanographie" (IMAGO),
Technopole Pointe du Diable, Plouzané, France
[5]LOCEAN, CNRS, IRD, Sorbonne Universités, MNHN, Paris, 75005, France

*Correspondence to*: Marie-Hélène Radenac (marie-helene.radenac@legos.obs-mip.fr)

**Abstract.** Ocean color observations show semiannual variations of chlorophyll in the Atlantic cold tongue with a main bloom in boreal summer and a secondary bloom in December. In this study, ocean color and in situ measurements, and a
coupled physical-biogeochemical model are used to investigate the processes that drive this variability. Results show that the main phytoplankton bloom in July-August is driven by a strong vertical supply of nitrate in May-July and the secondary bloom in December is driven by a shorter and moderate supply in November. The upper ocean nitrate balance is analyzed and shows that vertical advection controls the nitrate input in the equatorial euphotic layer and that vertical diffusion and meridional advection are key in extending and shaping the bloom off equator. Horizontal advection mostly acts to bring
nitrate low water below the mixed layer. Our results also give insights on the influence of intraseasonal processes in these exchanges. Observations and model show that the Equatorial Undercurrent brings low-nitrate water (relatively to off-equatorial surrounding waters) but still rich enough to enhance the cold tongue productivity.

## 1. Introduction

The Atlantic equatorial upwelling is a region of enhanced biological production mainly driven by nitrate supply (Voituriez
and Herbland, 1977; Loukos and Mémery, 1999). There, the availability of nutrients affects the equatorial ecosystem from primary production to high trophic levels and $CO_2$ fluxes (Hisard, 1973; Voituriez and Herbland, 1977; Oudot and Morin, 1987; Loukos and Mémery, 1999; Ménard et al., 2000; Christian and Murtugudde, 2003; Lefèvre, 2009). Early in situ measurements in the equatorial Atlantic (Hisard, 1973; Voituriez and Herbland, 1977) evidenced two seasons with different physical and biogeochemical conditions: i) a warm and low productive season in winter and spring with a nitrate depleted
surface layer and a chlorophyll maximum located near the top of the nitracline; ii) a cool and productive season in summer

---
[*] Deceased





and fall characterized by efficient vertical processes that bring cold and nitrate rich water supporting the phytoplankton growth in the euphotic layer. The advent of ocean color satellite measurements has made the monitoring of phytoplankton blooms possible and changed our vision of the equatorial variability. Using one year (March 1979-February 1980) of measurements from the Coastal Zone Color Scanner (CZCS), Monger et al. (1997) showed higher chlorophyll value (more than 1 mg m$^{-3}$) in October-December than in summer near 10° W. In contrast, during the first year of the Sea-viewing Wide Field-of-view Sensor (SeaWiFS), a bloom was observed between May and September and the October-December chlorophyll values were low (Signorini et al., 1999). This suggests large interannual fluctuations of the equatorial productivity. Nevertheless, a semiannual cycle of surface chlorophyll emerges as illustrated by the ocean color archive for the period 1998-2016 (Fig. 1a). This seasonal cycle is characterized by a primary chlorophyll bloom in July-August between 20° W and 5° W and a shorter and weaker second bloom in December (Perez et al., 2005; Grodsky et al., 2008; Jouanno et al., 2011a). Strong similarities between this seasonal cycle and the seasonal cycle of sea surface temperature (SST) suggest that the same physical processes could control the supply of cool and nutrient-rich waters into the euphotic layer (Hisard, 1973; Oudot and Morin, 1987; Grodsky et al., 2008; Jouanno et al., 2011a).

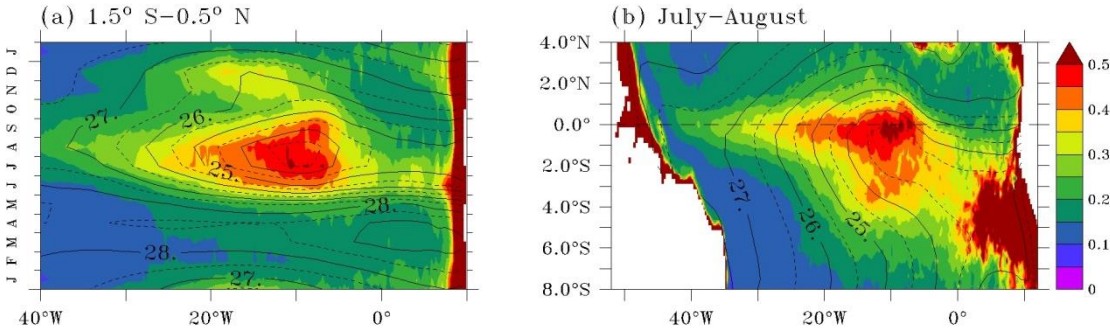

**Figure 1: Seasonal cycles averaged in 1.5° S-0.5° N (a) and mean surface distributions in July-August (b) of satellite chlorophyll (mg m$^{-3}$; colors) and observed SST (° C; contours). The SST contour interval is 0.5° C. Chlorophyll and SST climatologies are calculated over 1998-2015. See the data section for origin of data.**

Investigations of the link between physical processes and biological production in the equatorial Atlantic were conducted using in situ measurements during oceanographic cruises since the 60s and satellite measurements. The role of upwelling, vertical mixing, and variation of the depth of the thermocline/nitracline has been raised to explain the seasonal surface nitrate and chlorophyll increase. Hisard (1973) proposed that nutrient enrichment at 5° W is mainly driven by the equatorial divergence in summer and persists until fall because of enhanced vertical mixing. The enhancement of the vertical mixing during the cold season was associated with the strong vertical shear between the intensified South Equatorial Current (SEC) and the shallower Equatorial Undercurrent (EUC) by Voituriez and Herbland (1977). Considering oxygen and salinity distributions, Voituriez (1983) dismissed the influence of vertical mixing and emphasized the role of the thermocline/nitracline uplift. Oudot and Morin (1987) suggested that the equatorial divergence drove the summer nitrate enrichment and that its persistence until fall was supported by vertical mixing above the EUC core whose nitrate concentration increased because of the nitracline uplift. Monger et al. (1997) proposed that upwelling was the driving





mechanism of the summer and fall nitrate increase and that its efficiency was modulated by the relative depths of the EUC

and nitracline. Grodsky et al. (2008) stressed the role of the equatorial upwelling combined with the shoaling of the nitracline.

Few model based studies have addressed the influence of the ocean dynamics variability on the nutrient variability in the equatorial Atlantic. Loukos and Mémery (1999) used an offline nitrate transport model to examine the processes that drive

nitrate to the surface. In their two year simulation, surface nitrate concentration and biological production are more elevated in summer and decrease afterwards although they remain higher in fall-early winter than in spring. In summer, nitrate is brought to the EUC and euphotic layer through vertical advection and reaches the surface through vertical diffusion. Christian and Murtugudde (2003) ran a 50 year long coupled physical-biogeochemical model and underlined the influence of the relative depth between the nitracline and the upwelling core on the nitrate variations. In spring, the surface nitrate is at its

lowest because the upwelling is weak and located above the nitracline. In contrast, surface nitrate peaks in summer when water is upwelled from subsurface in response to the basin wide tilt of the thermocline/nitracline. More recently, Jouanno et al. (2011a) related processes responsible for SST changes to the observed chlorophyll changes. They highlighted the semiannual cycle of vertical mixing above the EUC core driven by the semiannual variation of the SEC. Maximum vertical mixing and surface cooling occur concurrently in summer while the impact of vertical mixing can be strongly damped by

air-sea heat fluxes during the secondary cooling in November-December. Because such a constraint does not exist for surface chlorophyll, intensified vertical mixing and surface chlorophyll peak simultaneously in summer and in November-December.

The impact of tropical instability waves (TIW) on the ecosystem of the Atlantic cold tongue was proposed by Morlière et al.

(1984) and Menkes et al. (2002). Although there is a debate about the TIW influence on the nutrient budget in the equatorial Pacific (Strutton et al., 2001; Gorgues et al., 2005), no such study is available in the equatorial Atlantic where TIW dominate the intraseasonal variability in the western and central basins and wind forced waves dominate in the east (Athié and Marin, 2008).

This study was motivated by observation of the nitrate vertical patterns during the low productive and productive seasons by repeated in situ measurements along 10° W acquired during recent cruises and their link with the semiannual variability of chlorophyll observed by ocean color satellites. Because cruise sampling prevents from studying an entire seasonal cycle, a coupled physical-biogeochemical simulation is used to complement the nitrate and chlorophyll seasonal cycles and to investigate the processes driving this seasonality. The datasets we use and the coupled physical-biogeochemical model are

described in Sect. 2. Previous studies have shown that vertical processes (equatorial divergence, vertical mixing, and vertical motion of the nitracline) are involved in setting the seasonal cycles of surface nitrate and chlorophyll. However, it is not clear how these vertical processes combine with horizontal processes to drive the bloom properties in terms of spatial extent





and duration. This issue is investigated by analyzing the model seasonal nitrate budget (Sect. 3). The role of the variation of nitrate concentration in the EUC on the nitrate budget in the euphotic layer and the impact of transient processes such as TIW and wind forced waves are discussed in Sect. 4. Concluding remarks are presented in Sect. 5.

## 2. In situ and satellite observations

### 2.1. Data sets

We use in situ nitrate, chlorophyll, and acoustic Doppler current profiler (ADCP) measurements collected during repeated transects along 10° W (Table 1) as part of the PIRATA (Prediction and Research Moored Array in the Tropical Atlantic; Servain et al., 1998; Bourlès et al., 2008; 2019) and EGEE (Étude de la Circulation Océanique et de sa Variabilité dans le Golfe de Guinée; Bourlès et al., 2007) programs. All these data, along with information on their acquisition and treatment, are available through their DOI (Bourlès, 1997; Bourlès et al., 2018a, 2018c). The analysis is based on 13 transects with nitrate measurements between 2004 and 2014 and four transects with chlorophyll measurement corresponding to the most recent French PIRATA cruises from 2011 to 2014. If we simply consider that upwelling conditions prevail when nitrate concentration larger than 1 μmol l$^{-1}$ is measured in the upper 10 m between 2° S and 1° N, only two cruises fulfill these conditions (June 2005 and July 2009). Note that there are no chlorophyll measurements during the boreal summer upwelling period.

|  | dates | NO$_3$ | Chl | U |
|---|---|---|---|---|
| FR12 | February 2004 | × |  | × |
| FR14-EGEE1 | June 2005 | × |  | × |
| EGEE2 | September 2005 | × |  | × |
| FR15-EGEE3 | June 2006 | × |  | × |
| EGEE4 | November 2006 | × |  |  |
| FR17-EGEE5 | June 2007 | × |  | × |
| EGEE6 | September 2007 | × |  |  |
| FR19 | July 2009 | × |  |  |
| FR20 | September 2010 | × |  |  |
| FR21 | May 2011 | × | × | × |
| FR22 | April 2012 | × | × | × |
| FR23 | May 2013 | × | × | × |
| FR24 | April 2014 | × | × | × |





**Table 1: list of PIRATA (FR) and EGEE transects along 10° W and availability of NO$_3$, chlorophyll, and ADCP zonal current**
**measurements.**

Observations from a PIRATA ocean-atmosphere interaction mooring and an ADCP mooring maintained at 10° W-0° N (Bourlès et al., 2018b) are also analyzed. We use monthly temperature measurements available since September 1997 at 1, 5, 10, 20, 40, 60, 80, 100, 120, 140, 180, 300 and 500 m depth, and daily ADCP current profiles available every 5 m from 15 m to about 300 m depth between December 2001 and March 2017.


The climatology of surface chlorophyll is calculated from chlorophyll estimates at 25 km horizontal resolution of the monthly GlobColour merged product obtained from different sensors and using the GSM (Garver, Siegel, Maritorena) model described in Maritorena et al. (2010). Sensors are Medium Resolution Imaging Spectrometer (MERIS), SeaWiFS, Moderate Resolution Imaging Spectroradiometer (MODIS)/Aqua, and Visible and Infrared Imager/Radiometer Suite (VIIRS) when 120 available.

The SST climatology is derived from the TropFlux dataset (Praveen Kumar et al. 2012). We use monthly SST maps from 1979 to 2016 at $1 \times 1$ degree resolution between 30° S and 30° N.

## 2.2. Observed seasonal cycles

Correspondences between spatial patterns and seasonal cycles of the surface chlorophyll and those of SST are illustrated in Fig. 1. In July-August, when the cold tongue expansion is the largest, the distribution of surface chlorophyll mirrors the SST distribution (Fig. 1b). The minimum SST and maximum surface chlorophyll coincide and are located south of the equator between 20° W and 5° W. Chlorophyll and SST gradients are sharper on the northern side of the cold tongue than on the southern side. The surface chlorophyll value starts to increase in May (Fig. 1a) and chlorophyll maximum and SST minimum 130 are found in July-August and in December. The December peak is better defined with chlorophyll than with temperature. The surface chlorophyll is at its minimum in spring and a secondary minimum occurs in October.

Vertical sections of nitrate, chlorophyll, and zonal current along 10° W measured during the PIRATA cruises and averaged separately in low productive and productive seasons are shown in Fig. 2a-c. Results are close to the distributions during the 135 cold and warm seasons described along 4° W in the 70s and 80s (Voituriez and Herbland, 1977; Oudot, 1983; Monger et al., 1997) and along 10° W during the June and September 2005 EGEE cruises (Nubi et al., 2016). During the warm and low productive season, the low chlorophyll (Fig. 2b) and nitrate depleted (Fig. 2a) layer extends from the surface to 30 m between 1° N and 5° S and deepens southward. Below, a nitracline ridge is observed between 2° S and 5° S. The deep chlorophyll maximum (DCM) is located in the upper nitracline and intensifies between 5° S and 2° N. During the cold and 140 productive season, the nitracline is uplifted and nitrate reaches the surface (Fig. 2c). The EUC transports water with low-nitrate concentration compared to off-equatorial waters at the same depth (Fig. 2a, 2c; Oudot, 1983), for both seasons.

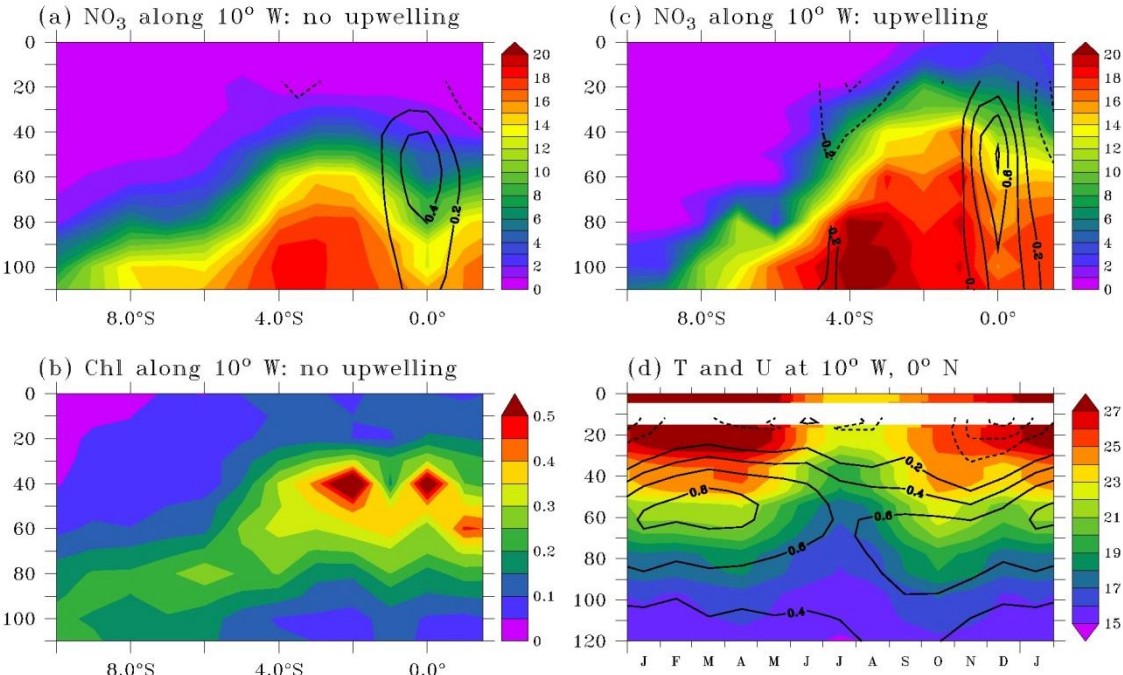

**Figure 2: (a, c)** Observed nitrate (µmol l⁻¹; a, c) and chlorophyll (mg m⁻³; b) distributions along 10°W during low productive (a, b) and productive (c) conditions. Zonal velocity is overlaid on nitrate distribution. **(d)** Seasonal cycles of temperature (colors; ° C) and zonal current (contours; m s⁻¹) at the 10° W, 0° N mooring. Velocity contour interval is 0.2 m s⁻¹; the 0 m s⁻¹contour has been removed.

Previous studies have shown that the location of the thermocline/nitracline relative to the EUC depth impacted the efficiency of the upwelling (Monger et al., 1997; Christian and Murtugudde, 2003). Figure 2d illustrates the seasonal vertical excursions of the thermocline and EUC as deduced from the temperature and zonal current profiles measured at the 10° W, 0° N PIRATA mooring. Seasonal variations of the depth of the EUC core are small while the thermocline depth shows larger vertical movements. The thermocline is about 20 m below the EUC core in April and about 30 m above in August, leading to variations in the properties of the EUC water. The temperature is colder in the EUC in August than during spring. Likewise, nitrate is more elevated in the EUC in August than during spring (Oudot and Morin, 1987), as expected from the strong relationship between nitrate and temperature in the nitracline of the cold tongue all year long (Voituriez and Herbland, 1984).

## 3. Coupled physical-biogeochemical simulation

### 3.1. Model description

A coupled simulation is used to describe the nitrate seasonal cycle and the seasonal nitrate budget in the mixed and euphotic layers. The physical component of the simulation is based on the NEMO (Nucleus for European Modeling of the Ocean;





Madec et al., 2016) numerical code. We use the regional configuration described in Hernandez et al. (2016; 2017) that covers the tropical Atlantic between 35° S and 35° N and from 100° W to 15° E. The resolution of the horizontal grid is ¼° and there are 75 vertical levels, 24 of which are in the upper 100 m layer. The depth interval ranges from 1 m at the surface to about 10 m at 100 m depth. The MERCATOR global reanalysis GLORYS2V4 (Storto et al., 2018) is used to force the model at the lateral boundaries. Interannual atmospheric fluxes of momentum, heat, and freshwater are derived from the DFS5.2 product (Dussin et al., 2016) using bulk formulae from Large and Yeager (2009).

The physical model is coupled to the PISCES (Pelagic Interaction Scheme for Carbon and Ecosystem Studies) biogeochemical model (Aumont et al., 2015) that simulates the biological production and the biogeochemical cycles of carbon, nitrogen, phosphorus, silica, and iron. Two phytoplankton classes (nanophytoplankton and diatoms) differ by their silicate and iron requirements. The two zooplankton compartments (nanozooplankton and mesozooplankton) feed on the two phytoplankton classes. The model also includes three non-living compartments (dissolved organic matter, small and large sinking particles). The biogeochemical model is initialized and forced at the lateral boundaries with dissolved inorganic carbon, dissolved organic carbon, alkalinity, and iron obtained from stabilized climatological 3-D fields of the global standard configuration ORCA2 (Aumont and Bopp, 2006), and nitrate, phosphate, silicate, and dissolved oxygen from the World Ocean Atlas observation database (WOA; Garcia et al., 2010).

The model is integrated from 1993 to 2015 and monthly averages for the period 1995 to 2015 are analyzed. Such short spin-up is justified by the fast adjustment of the equatorial dynamics and the main focus of the study which is on the upper ocean variability.

The three-dimension nitrate balance solved in the model reads as follows:

$$\frac{\partial NO_3}{\partial t} = -u\frac{\partial NO_3}{\partial x} - v\frac{\partial NO_3}{\partial y} - w\frac{\partial NO_3}{\partial z} + D_l(NO_3) + \frac{\partial}{\partial z}\left(K_z\frac{\partial NO_3}{\partial z}\right) + SMS \tag{1}$$

in which $NO_3$ is the model nitrate concentration, (u, v, w) are the velocity components, $D_l(NO_3)$ is the lateral diffusion operator, and $K_z$ is the vertical diffusion coefficient for tracers. The first three terms on the right-hand side are the zonal, meridional, and vertical advections; the fourth and fifth terms are the lateral and vertical diffusions. The last term, called "source minus sink" (SMS), is the nitrate change rate due to biogeochemical processes which include uptake by nanophytoplankton and diatoms, nitrification, denitrification, and nitrogen fixation. The different terms are computed on-line and averaged over one month periods.

We estimate the low and high frequency contributions to the advection terms by separating off line each advection term into low frequency and submonthly components:

$$-\overline{u\frac{\partial NO_3}{\partial x}} = -\bar{u}\frac{\overline{\partial NO_3}}{\partial x} - \overline{u'\frac{\partial NO_3'}{\partial x}} \tag{2}$$





The left hand side term is the monthly average of zonal advection. On the right hand side, the first term is the monthly zonal advection calculated from monthly averages of zonal current ($\bar{u}$) and nitrate concentrations ($\overline{NO3}$). The advection by

submonthly currents (last term), which in this region may include inertia-gravity waves, mixed Rossby-gravity waves, Kelvin waves, and eddies or tropical instability waves (e.g. Athié et al. 2009; Jouanno et al. 2013) is given by the residue between the total and mean zonal advection. Meridional and vertical advections are decomposed in the same way. Such decomposition has been used to estimate the eddy contribution to SST budget in the Pacific mixed layer (Vialard et al., 2001) and oxygen advection in the Arabian Sea (Resplandy et al., 2012).


We use the method described in Vialard and Delecluse (1998) to investigate nitrate budgets in the mixed layer and in the euphotic layer. The entrainment term appears when integrating the nitrate budget over a time-varying layer. It is calculated as the residual of the nitrate budget, as done in Vialard and Delecluse (1998) for temperature. The mixed layer depth is computed as the depth where the density is 0.03 kg m$^{-3}$ higher than the 10 m density (de Boyer Montégut et al., 2004) and

the depth of the euphotic layer is the depth where the surface photosynthetically available radiation (PAR) is reduced to 1 % (Morel and Berthon, 1989). The contributions of entrainment and lateral diffusion to the nitrate budgets in both layers are weak and are not shown.

### 3.2. Evaluation of the modeled seasonal cycle

The climatology of the simulated surface chlorophyll calculated over the same period (1998-2015) than observations (Fig. 1)

is shown in Fig. 3. The model reproduces the pattern and semiannual variability of surface chlorophyll in the equatorial cold tongue. The simulated chlorophyll maximum is shifted about 5° east of the chlorophyll maxima observed by satellite. The model surface chlorophyll is also slightly higher than observed east of the equatorial chlorophyll maximum.

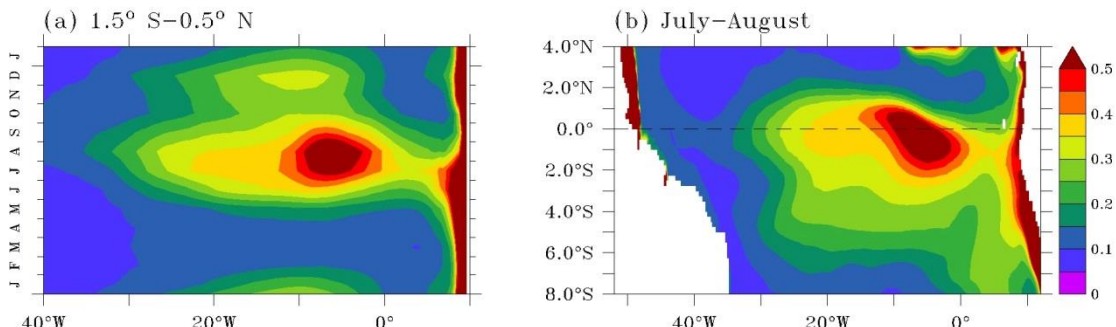

**Figure 3: Seasonal cycles averaged in 1.5° S-0.5° N (a) and mean surface distributions in July-August (b) of the simulated**

**chlorophyll (mg m$^{-3}$; colors). The climatology is calculated over 1998-2015.**

The meridional sections of modeled nitrate and chlorophyll along 10° W presented in Fig. 4a-c have been calculated using fields coincident with observed sections in Fig. 2a-c. The model properly reproduces the main features such as the nitracline uplift around 3° S and the low-nitrate signature of the EUC (Fig. 4a, c). However, the simulated nitrate has a positive bias





that can reach 5 µmol l⁻¹ in the nitracline in the 5° S-2° S region. In the equatorial zone, the model nitrate is slightly
overestimated (less than 1 µmol l⁻¹) above the 5 µmol l⁻¹ nitrate isoline (which is close to the 20° C isotherm depth) and
slightly underestimated (about 1 µmol l⁻¹) below. The nitrate depleted surface layer is about 10 m shallower in the simulation
than in the observations. The position of the simulated DCM in the upper nitracline is in agreement with observations while
its magnitude is more elevated by about 0.1 mg m⁻³ (Fig. 4b). Too elevated simulated chlorophyll is found up to the surface
where the concentration is about 0.15 mg m⁻³ at the equator instead of 0.1 mg m⁻³ in the observations.

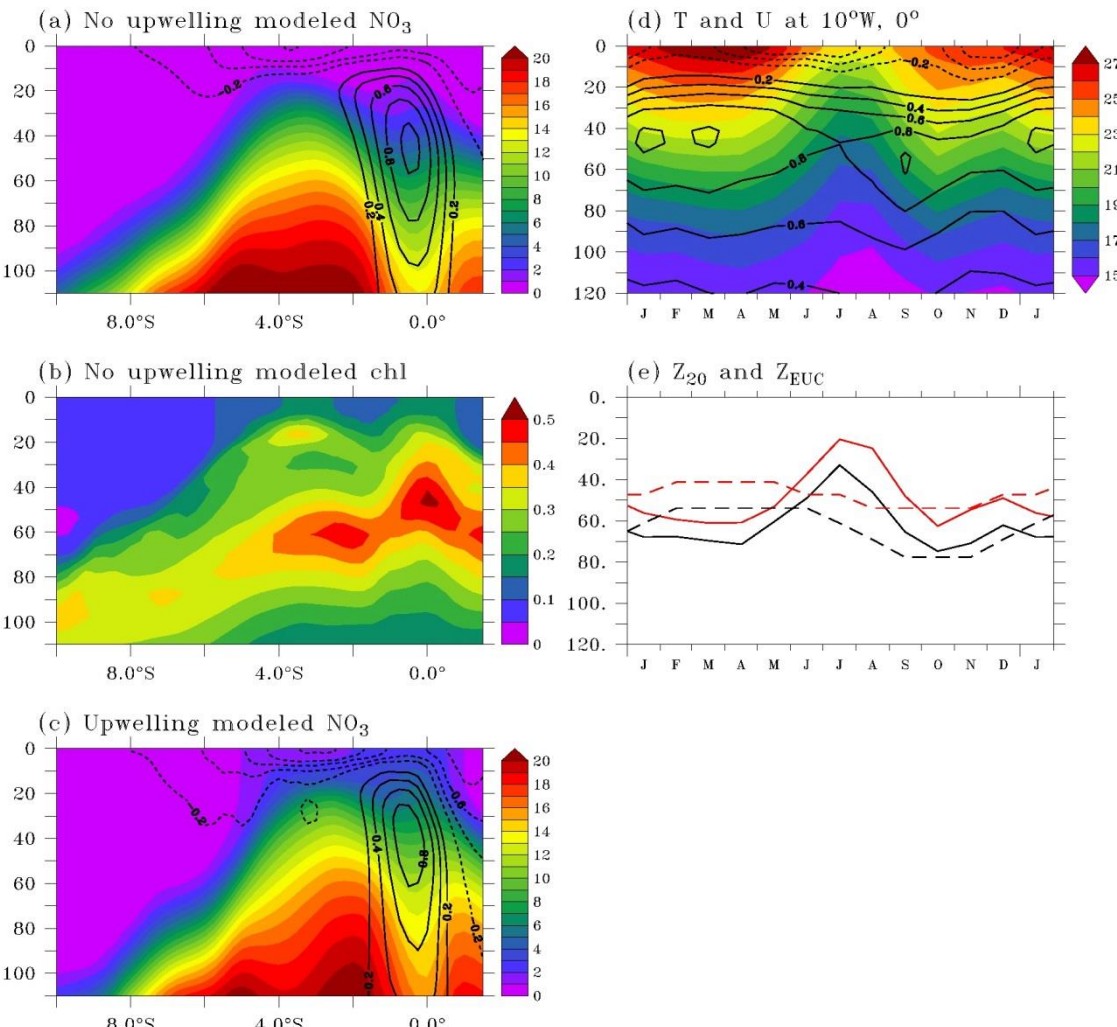


**Figure 4: (a, c) Simulated nitrate (µmol l⁻¹; a, c) and chlorophyll (mg m⁻³; b) distributions along 10° W during low productive (a, b)
and productive (c) conditions. Zonal velocity is overlaid on nitrate distribution. (d) Seasonal cycles of temperature (colors; ° C)
and zonal current (contours; m s⁻¹) at the 10° W, 0° N mooring. Velocity contour interval is 0.2 m s⁻¹; the 0 m s⁻¹ contour has been
removed. (e) Observed (black) and simulated (red) depths of the 20° C isotherm (full line) and of the EUC core (dashed line) at 10°**
**W, 0° N.**





Simulated profiles of temperature and zonal current coincident with available observed profiles (Fig. 2d) at the PIRATA mooring at 10° W, 0° N were used to calculate the climatology shown in Fig. 4d. The amplitude and phase of the seasonal cycles of modeled temperature and zonal current compare well, although the simulated temperature and current vertical structures are too shallow. The depths of the 20° C isotherm (Z20) and of the EUC core (ZEUC) are 12 and 16 m shallower

than observed, respectively (Fig. 4e). However, the relative position of Z20 and ZEUC is correctly reproduced. The simulated nitrate concentration at ZEUC (not shown) is less than 2 µmol l$^{-1}$ in spring and rises to nearly 9 µmol l$^{-1}$ in August, in agreement with observations at 4° W (Oudot and Morin, 1987). In the 20 m surface layer, the model well captures the weakening of the SEC in January-February and September-October (Okumura and Xie, 2006; Ding et al., 2009; Habasque and Herbert, 2018).

**4. The modeled nitrate seasonal cycle**

The good agreement between the observed and simulated patterns and seasonal variations of chlorophyll and nitrate makes the model a relevant tool to investigate the seasonal nitrate budget. In this section, processes driving the seasonal variations of nitrate are presented in the mixed layer, but also in the euphotic layer because the mixed layer budget cannot explain alone the surface productivity. We focus on the 20° W-5° W, 1.5° S-0.5° N region where the surface chlorophyll values are the

largest.

**4.1. Nitrate budget in the mixed layer**

In the equatorial Atlantic, the seasonal variations of chlorophyll are thought to be primarily related to seasonal variability of the nitrate input (Voituriez and Herbland, 1977; Loukos and Mémery, 1999). This is well illustrated by the seasonal cycle of nitrate built from 21 years of simulation (1995-2015) in Fig. 5a, that closely matches the seasonal variability of the model

(Fig. 3b) and satellite chlorophyll (Fig. 1a).

The mixed layer nitrate concentration at the equator shows large and coherent variations between 30° W and 0° E (Fig. 5a). This central equatorial variability does not seem to be directly connected with mixed layer nitrate input along the African coast, in agreement with the chlorophyll behavior in the equatorial Atlantic deduced from satellite data (Grodsky et al.,

2008). Four phases emerge from the nitrate seasonal evolution in the mixed layer of the central equatorial Atlantic: a nitrate increase between April and July, a decrease between August and October, a rapid increase in November, and a secondary decrease starting in December (Fig. 5b). This temporal pattern results from the imbalance between physical processes (Fig. 5h) that brings nitrate into the mixed layer and nitrate uptake by the biological activity (Fig. 5g). Unlike the negligible contribution of vertical advection in the temperature budget in the mixed layer of the equatorial Atlantic (Jouanno et al.,

2011b), both vertical advection (Fig. 5e) and vertical diffusion (Fig. 5f) contribute to nitrate inputs in the mixed layer. Horizontal advection (Fig. 5c, d) mainly acts to bring nitrate low waters to the cold tongue area.





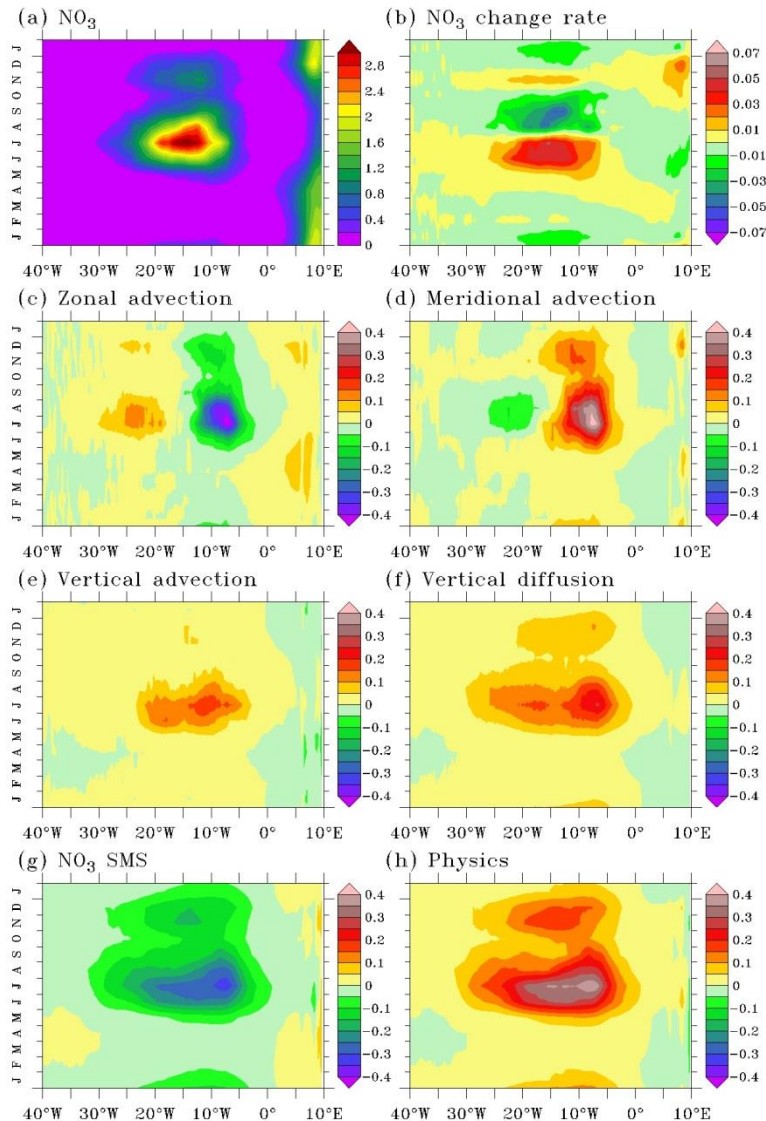

**Figure 5: Seasonal cycle of modeled (a) surface nitrate (µmol l⁻¹), (b) nitrate change rate, (c) zonal advection, (d) meridional advection, (e) vertical advection, (f) vertical diffusion, (g) nitrate source minus sink, and (h) physical processes averaged in 1.5° S-0.5° N in the mixed layer. Tendency units are µmol l⁻¹ day⁻¹. Note that the color scale of nitrate change rate is different from the color scale of other tendencies. Climatology has been calculated between 1995 and 2015.**

In Fig. 6, we show the regional distribution of the different terms of the nitrate balance in July, when the nitrate supply to the mixed layer by physical processes and nitrate uptake are at their maximum. During this period, the biological sink is not efficient enough to offset the physical supply and the surface nitrate (Fig. 6a) shows a maximum between 1.5° S and 0.5° N from 20° W to 5° W at the location of maximum vertical input through advection (Fig. 6e) and diffusion (Fig. 6f), and biological sink (Fig. 6g). Advection of nitrate poor water from the east (Fig. 6c) and meridional advection (Fig. 6d) counteract the vertical nitrate supply near the equator. Off the equator, meridional advection acts to spread the upwelled





nitrate rich water poleward along the northern boundary of the nitrate rich patch and, to a lesser extent, along the southern
boundary where the nitrate gradient is weaker. This may contribute to the meridional extension of the observed (Fig. 1b) and
model (Fig. 3a) chlorophyll distribution.

The scenario leading to the December secondary nitrate maximum in the mixed layer is close to the boreal summer nitrate
evolution, except that the duration of the processes is shorter (about one month long), their magnitudes are weaker, and they
span a narrower longitudinal range (Fig. 5). Between the summer and December nitrate maxima, vertical processes strongly
decrease (Fig. 5c, f) and sustain less nitrate supply in the mixed layer, allowing the biological sink (Fig. 5g) to prevail over
the physical input (Fig. 5h).

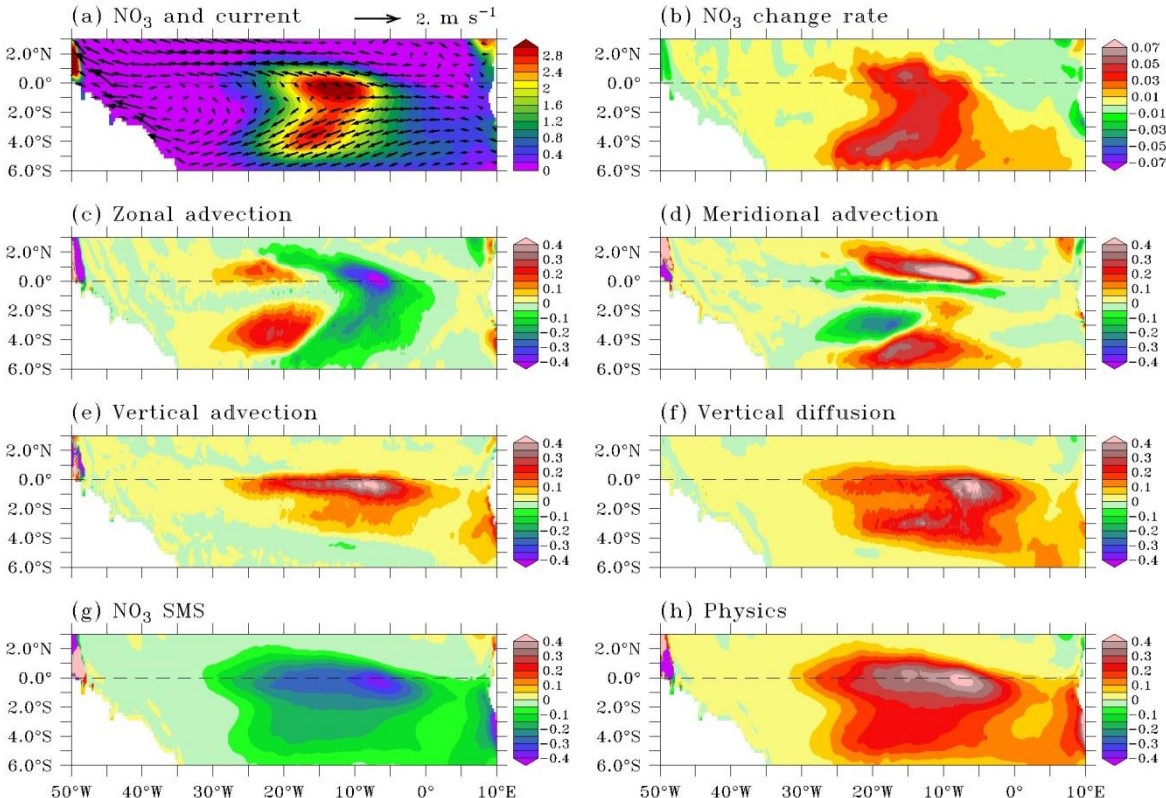

**Figure 6: Maps of (a) nitrate, (b) nitrate change rate, (c) zonal advection, (d) meridional advection, (e) vertical advection, (f)
vertical diffusion, (g) nitrate source minus sink, and (h) physical processes averaged in the mixed layer in July. The mean current
in the mixed layer is superimposed in (a). Note that color scale in (b) is different from color scale in (c-h). Nitrate units are µmol l$^{-1}$
and tendency units are µmol l$^{-1}$ day$^{-1}$.**

### 4.2. Nitrate budget in the euphotic layer

In this section, we examine how, in addition to processes in the mixed layer, variations of nitrate below the mixed layer
impact variations of surface nitrate and, in turn, of chlorophyll. Also, variations in the euphotic layer where the biological





production takes place allow better explaining the transition between the low productive and productive seasons. The seasonal cycles of chlorophyll, nitrate, and of the main processes involved in nitrate change in the 20° W-5° W, 1.5° S-0.5° N region from the surface to 80 m are shown in Fig. 7. Depths of the mixed layer, of the euphotic layer, and of the EUC core are overlaid. The depth of the thermocline core is represented by the 20° C isotherm depth. The separated low frequency and submonthly contributions to the advection terms are presented in Fig. 8.

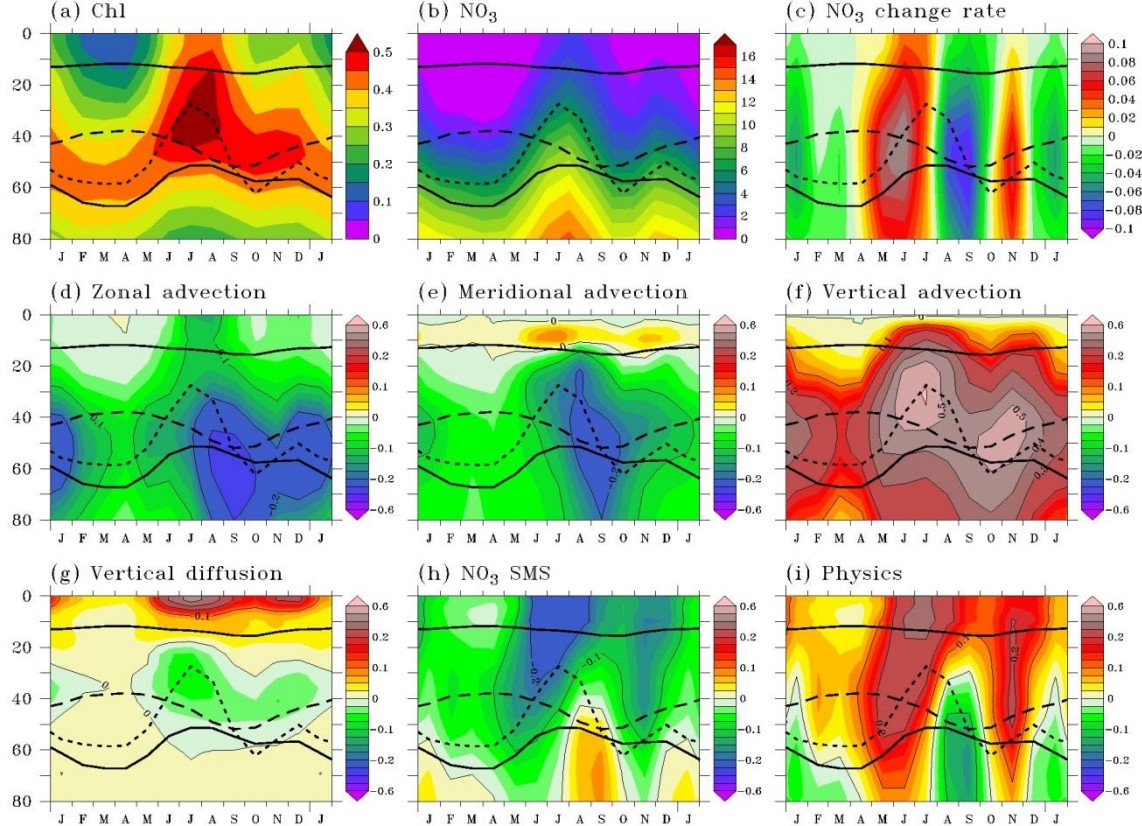

**Figure 7: Seasonal cycle of vertical profiles of (a) chlorophyll, (b) nitrate, (c) nitrate change rate, (d) zonal advection, (e) meridional advection, (f) vertical advection, (g) vertical diffusion, (h) nitrate source minus sink, and (i) physical processes averaged in 20° W-5° W, 1.5° S-0.5° N. Chlorophyll units are mg m⁻³, nitrate units are µmol l⁻¹, and tendency units are µmol l⁻¹ day⁻¹.**

**Figure 7: Seasonal cycle of vertical profiles of (a) chlorophyll, (b) nitrate, (c) nitrate change rate, (d) zonal advection, (e) meridional advection, (f) vertical advection, (g) vertical diffusion, (h) nitrate source minus sink, and (i) physical processes averaged in 20° W-5° W, 1.5° S-0.5° N. Chlorophyll units are mg m$^{-3}$, nitrate units are µmol l$^{-1}$, and tendency units are µmol l$^{-1}$ day$^{-1}$. Tendency contours are every 0.1 µmol l$^{-1}$ day$^{-1}$. The depths of the mixed layer (upper solid line), of the euphotic layer (lower solid line), of the EUC core (dashed line), and of the 20° C isotherm (dotted line) are indicated. Note that color scale in (c) is different from color scale in (d-i).**

The semiannual cycle of chlorophyll described in the mixed layer is also visible in the entire euphotic layer (Fig. 7a). The seasonal cycle of the depth of the simulated DCM is in agreement with observations (Monger et al., 1997). It is located near the thermocline core between 50 and 60 m in spring, raises toward the surface at the same time than the thermocline core in summer, sinks in early fall, and raises again in November. In February-April, chlorophyll values are low in the nitrate depleted surface layer as in oligotrophic ecosystems. The semiannual variations of chlorophyll in the euphotic layer are closely associated with semiannual variations of nitrate (Fig. 7b).





The nitrate change rate is maximum at the base of the euphotic layer near the EUC core (Fig. 7c). Its semiannual cycle can

be seen as interplays between the physical supply (Fig. 7i) and the biological sink (Fig. 7h). Physical processes mostly bring nitrate into the euphotic layer with maximum input in the mixed layer in May-August and below the mixed layer in November. In contrast, nitrate is consumed by the biological activity in the euphotic layer and remineralized below. Physical supply is stronger than biological loss during the main peak of the nitrate change rate in May-July. During the short second peak in November; biological losses prevail over physical supply in August-October and in December-January. The nitrate

supply in November suggests that the observed and simulated elevated chlorophyll values in December result from a second chlorophyll bloom and not from a persistence of elevated nitrate and chlorophyll concentrations following the summer bloom (Hisard, 1973; Oudot and Morin, 1987).

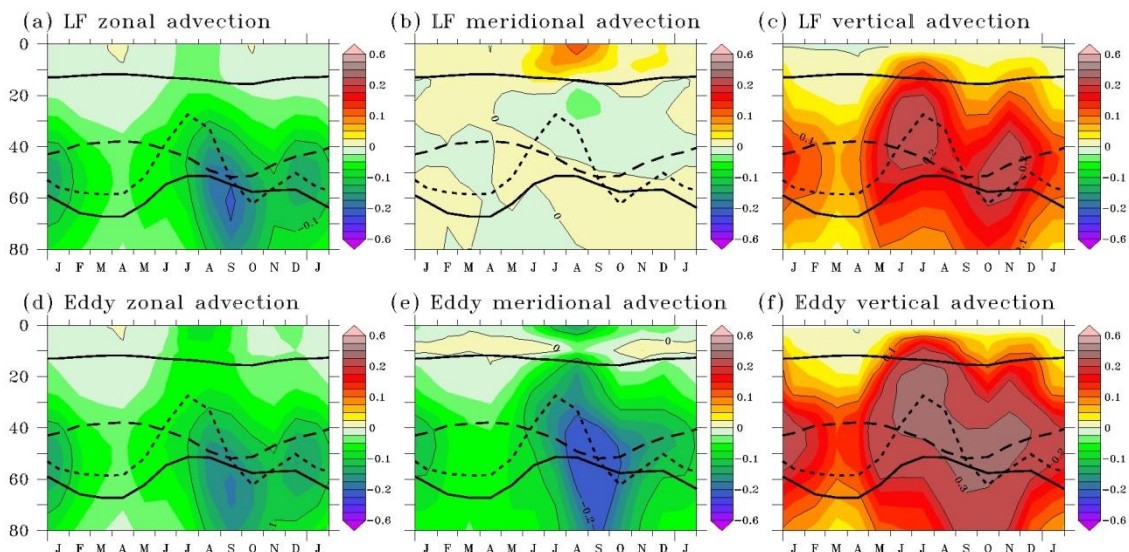

**Figure 8: Seasonal cycle of vertical profiles of low frequency (LF; upper panels) and eddy (lower panels) zonal advection (a, d),**
**meridional advection (b, e), and vertical advection (c, f). Tendency contours are every 0.1 µmol l⁻¹ day⁻¹. The depths of the mixed layer (upper solid line), of the euphotic layer (lower solid line), of the EUC core (dashed line), and of the 20° C isotherm (dotted line) are indicated.**

Vertical advection always brings nitrate into the euphotic layer (Fig. 7f). It drives the main nitrate increase in May-July and the secondary one in November when easterly winds strengthen. The maximum vertical advection is located near the layer of

maximum vertical nitrate gradient, close to the depth of the 20° C isotherm, and it occurs when the vertical velocity is strong (July and November). Vertical advection of nitrate rich water at the base of the mixed layer favors the intensified vertical diffusion in summer and November-December (Fig. 7g), with an acceleration of the SEC which increases the vertical shear with the EUC and, in turn, increases the vertical mixing in the mixed layer (Jouanno et al., 2011b). Both the low frequency (Fig. 8c) and eddy (Fig. 8f) advections contribute to the nitrate supply through vertical advection, especially in the upper

EUC between June and December. The eddy advection is more sustained than the low frequency advection.

Horizontal advection (Fig. 7c, d) removes nitrate all year long. It drives the strong nitrate loss in August-September and the lesser one in December-January (Fig. 7b) when the contributions of both zonal (Fig. 7c) and meridional (Fig. 7d) advections are the largest. The contribution of the low frequency advection (Fig. 8a) compares to that of the eddy advection (Fig. 8d). In the mixed layer, the low frequency advection of nitrate poor water from the east is the largest where the zonal nitrate gradient is the strongest. Below the mixed layer, negative low frequency zonal advection indicates the eastward transport of low-nitrate water by the EUC. Meridional advection is a weak source of nitrate in the mixed layer while it significantly contributes to the nitrate decrease below (Fig. 7d). The low frequency advection (Fig. 8b) reveals the influence of the equatorial cell: the northward transport of nitrate rich upwelled water dominates the meridional advection in the mixed layer on average in the 20° W-5° W, 1.5° S-0.5° N region. Below the mixed layer, nitrate poor water is advected from the north by the low frequency southward component of the subsurface current (Perez et al., 2014). Below the mixed layer, the eddy signal (Fig. 8e) controls the meridional advection.

## 5. Discussion

Observations and the model used in this study show semiannual cycles of chlorophyll and nitrate. The model further shows that they are sustained by semiannual variations of processes in the euphotic layer. Changes of nitrate properties in the EUC and intraseasonal processes are involved in shaping the seasonal cycle of nitrate supply and losses.

### 5.1. Variability of nitrate in the equatorial undercurrent

Upwelled water in the central basin originates in the upper part of the EUC (Fig. 7f). The EUC waters mainly originate in the very oligotrophic ecosystem of the south subtropical gyre (Oudot, 1983; Blanke et al., 2002; Hazeleger et al., 2003; Aiken et al., 2017). Waters are transported westward, feed the North Brazil Undercurrent (NBUC) and are entrained within the North Brazil Current retroflection before entering the EUC. A small fraction of water also originates in the North Equatorial Current (Bourlès et al., 1999; Hazeleger et al., 2003). Therefore, this may explain that water transported eastward by the EUC has relatively low-nitrate concentrations compared to nearby north and south water masses (Fig. 3a, c) in agreement with in situ measurements (Oudot, 1983). This relatively low-nitrate water is upwelled toward the surface layer along the equator.

Seasonal changes of the nitrate concentration in the EUC in the central equatorial basin are closely related to the seasonal nitracline shoaling (Oudot and Morin, 1987). Its semiannual cycle follows the basin wide adjustment of the thermocline to the wind forcing via interactions between wind forced Kelvin waves and boundary reflected Rossby waves (Merle, 1980; Ding et al., 2009). It conditions the depth of the thermocline and associated nitracline that varies from 60 m in spring to about 20 m in July-August while the upwelling core remains in the upper part of the EUC, at 20-30 m, all year long (Fig. 4e).





The smallest vertical supply (Fig. 7f) occurs when the nitracline is well below the weak upwelling core in spring. In May-July and November, the vertical velocity is strong and the nitracline gets closer to the upwelling core, allowing vertical advection to increase.


The annual shoaling of the thermocline in the western basin associated with the semiannual shoaling in the central basin leads to a strong zonal slope of the thermocline depth in July-September and December-January (Ding et al., 2009) and also of the nitracline. During these periods of time, the resulting strongly negative zonal advection (Fig. 7d) in the EUC underlines the efficiency of the EUC in reducing the local nitrate concentrations. Nitrate removal by zonal advection in the
EUC contributes to decrease the vertical nitrate gradient, which, associated with a reduced vertical velocity, leads to moderate vertical nitrate supply in August-September. At that time of the year, physical processes drive reduced nitrate supply in the upper part of the euphotic layer and nitrate removal in its deeper part (Fig. 7i).

The seasonal nitrate supply in the center of the equatorial Atlantic is supported by vertical processes and strongly modulated
by losses through horizontal advection in the EUC linked to the semiannual thermocline uplift of the nitracline. Variations of the nitrate concentration in the source waters of the NBUC may be another driver of nitrate variations in the EUC as suggested by White (2015) who finds that variations of temperature in the NBUC contribute to variations of the cold tongue SST 6 to 8 months later. Changes along the EUC pathway (meridional circulation in the tropical cells, elevation of the nitracline in the west, intraseasonal processes) may also impact horizontal and vertical nitrate gradient and the rates of
supply and removal of nitrate in the central equatorial Atlantic. This will deserve further attention.

### 5.2. Intraseasonal processes

On average in the 20° W-5° W, 1.5° S-0.5° N region, our model results show that horizontal eddy advection is responsible for nitrate decrease, especially in August-September, and that vertical eddy advection supplies the euphotic layer with nitrate. The intraseasonal nitrate variations may have the same origins as temperature modulations observed at periods
between 10 and 50 days in the cold tongue (Marin et al., 2009; de Coëtlogon et al., 2010; Jouanno et al., 2013; Herbert and Bourlès, 2018). TIW are observed west of 10° W at periods between 20 and 50 days (Jochum et al., 2004; Athié and Marin, 2008; Jouanno et al., 2013). East of 10° E, the impacts of wind forced equatorial waves superimpose at different frequencies (Houghton and Colin, 1987; Athié et al., 2008; de Coëtlogon et al., 2010; Jouanno et al., 2013; Herbert and Bourlès, 2018): Kelvin waves at periods between 25 and 40 days, mixed Rossby-gravity waves between 15 and 20 days, and inertia-gravity
waves between 5 and 11 days.

Considering the upper 20 m in the equatorial Atlantic, Jochum et al. (2004) found that the annual meridional advective heat flux associated with TIW was nearly offset by the vertical advective heat flux. In contrast, Peter et al. (2006) attributed the warming in the mixed layer induced by eddy horizontal advection between 30° W and 5° W to TIW because of strong





southward heat transport. In the Pacific Ocean, the compensation between the TIW horizontal and vertical heat advection in
the mixed layer was also suggested by Vialard et al. (2001). However, Menkes et al. (2006) found that the TIW vertical
advection was low and that the TIW effect was to warm the Pacific cold tongue in the upper 200 m. Mixed Rossby-gravity
waves, inertia-gravity waves, and Kelvin waves are believed to contribute to cooling the Atlantic cold tongue through both
northward advection of cold tongue water and vertical mixing (Houghton and Colin, 1987; Marin et al., 2009; Jouanno et al.,
2013), although no calculations of the heat budget were done.

The coincidence of high chlorophyll concentrations with meridional oscillations of currents associated with an anticyclonic
eddy observed during a summer cruise in the equatorial Atlantic (Morlière et al., 1994) strongly suggests that TIW may also
influence ecosystems. This was further settled with synoptic observations of physical (temperature, salinity, current) and
ecosystem (nitrate, chlorophyll, zooplankton, micronekton) tracers in a tropical instability vortex (Menkes et al., 2002): their
horizontal and vertical structures were highly coherent. As for the heat budget, the impact of TIW on biological production is
debated, at least in the equatorial Pacific Ocean. Gorgues et al. (2005) show that the effect of TIW is to lower the chlorophyll
concentration near the equator because of the iron loss through horizontal advection exceeds iron supply by vertical
advection while Strutton et al. (2001) show that chlorophyll increases because of enhanced upwelling. As far as we know, no
study shows the possible impact of TIW and other intraseasonal waves on nitrate budget in the Atlantic Ocean.

The more elevated surface chlorophyll concentrations are found in the 20° W-5° W, 1.5° S-0.5° N zone which is affected by
TIW and Kelvin waves in the 20-50 day period range, and by mixed Rossby-gravity and inertia-gravity waves at higher
frequency. In this study, a one month threshold separates the eddy signal from the low frequency signal. So, the impacts of
mixed Rossby-gravity and inertia-gravity waves and part of the variability associated with TIW and Kelvin waves at periods
shorter than one month enters the eddy advection terms. The part of the TIW and Kelvin waves signal with longer periods is
included in the low frequency advection terms.

Several intraseasonal processes should contribute to the seasonal nitrate loss through eddy horizontal advection and nitrate
input through eddy vertical advection in the mixed layer and in the euphotic layer in the 20° W-5° W, 1.5° S-0.5° N region.
In this simulation, nitrate loss in the mixed layer west of 10° W is driven by eddy meridional advection and by eddy zonal
advection to a lesser extent (not shown). As the horizontal and vertical patterns of temperature and nitrate in a tropical
instability vortex are close (Menkes et al., 2002), the advection of nitrate anomaly by eddy zonal and meridional currents
could drive nitrate losses by eddy zonal and meridional advection in the same way as the advection of anomalous
temperature by anomalous currents drives a mixed layer warming close to the equator. By analogy with TIW induced
warming (Vialard et al., 2001; Peter et al., 2006; Menkes et al., 2006), TIW could be a strong contributor to the nitrate eddy
term. Nitrate loss by eddy horizontal advection is also consistent with iron loss associated with TIW in the equatorial Pacific
(Gorgues et al., 2005). East of 10° W, the nitrate removal through eddy horizontal advection is driven by eddy zonal

**Biogeosciences** Open Access
Discussions
EGU

advection while eddy meridional advection strongly decreases (not shown). Drawing again an analogy between temperature and nitrate, the nitrate decrease through zonal advection could be attributed to Kelvin waves. In contrast, no nitrate increase through meridional advection is simulated as would be expected from mixed Rossby-gravity, inertia-gravity, and Kelvin waves that cool the mixed layer. The conclusion on the nature of intraseasonal processes that affect the nitrate budget east of 10° W is not straightforward. One reason could be that the low frequency signal captures part of the Kelvin wave induced variability as there is no sharp cutoff at 30 days in the spectrum of Kelvin waves (Athié and Marin, 2008; Athié et al., 2009;

Jouanno et al., 2013). Another reason would be related to the different distribution of temperature and nitrate in the mixed layer because the nitrate concentration rapidly drops to zero east of 10° W while a temperature gradient persists in this simulation.

On an annual average, nitrate is supplied by intraseasonal advection because eddy induced vertical advection exceeds

horizontal advection. It represents a significant contribution to the nitrate budget in the central equatorial Atlantic: about 35% of the advective nitrate input in the mixed layer and about 45% in the euphotic layer. It differs from the overall warming contribution of TIW to the SST budget of the equatorial Pacific cold tongue showed by Menkes et al. (2006). This warming contribution reflects the impact of horizontal advection as TIW induced vertical advection is negligible. As far as horizontal eddy advection is concerned, the warming effect of zonal and meridional advections in the Pacific is consistent

with the nitrate removal by zonal and meridional advections in the Atlantic.

Although this crude qualitative comparison does not allow concluding about the role of the different intraseasonal processes, this simulation initially designed to study the large scale processes shows that they cannot totally explain the seasonal evolution of the nitrate budget. A dedicated study allowing better separating the large scale and eddying signals is needed in

order to identify the nature of intraseasonal processes at work and their impact on the seasonal nitrate budget in the Atlantic cold tongue area.

## 6. Conclusion

We described and analyzed the seasonal cycle of nitrate and the associated physical processes in the Atlantic cold tongue region using in situ and satellite data, and a coupled physical-biogeochemical simulation. The model reproduces the

horizontal and vertical patterns of chlorophyll observed in the studied area and its semiannual cycle. Nitrate required for the phytoplankton growth is supplied by vertical processes. The main supply period occurs from May to July and a secondary supply also occurs in November. In between, nitrate is removed by horizontal advection in August-September and during the secondary loss event in December-January. We draw attention to the potential roles of nitrate variations in the EUC and of intraseasonal processes on the seasonal nitrate budget.


Ding et al. (2009) put forward the presence of a basin mode that explains semiannual changes of SSH gradient. Our results show how the thermocline and nitracline uplift affects the zonal nitrate gradient in the EUC and so, how it influences nitrate removal by horizontal advection and then vertical supply. Changes of the nitrate concentration in the source water within the NBUC may also impact nitrate changes in the center of the basin. A dedicated study to nitrate variations in the EUC and
associated processes from the inflow in the western boundary current system to the equatorial upwelling region would contribute to better understand phytoplankton variations in the equatorial Atlantic.

Our results suggest that eddy horizontal advection acts to remove nitrate while eddy vertical advection feeds both the mixed and euphotic layers with nitrate. Overall, eddy advection brings nitrate into the mixed and euphotic layers in June-July and in
November-December. To our knowledge, there are no studies on the role of TIW and other intraseasonal processes on the equatorial Atlantic nitrate budget. This issue should be further investigated.

**Data availability.**

PIRATA data sets of cruise and mooring measurements are available through http://www.brest.ird.fr/pirata and their DOI. The ocean color products of the GlobColour project are available through http://globcolour.info. The TropFlux data are
archived at https://www.incois.gov.in/tropflux. Model results can be reproduced by using the ocean code nemo_v3_6 (http://forge. ipsl.jussieu.fr/nemo/wiki/Users). The DFS5.2 forcing set is available on the server http://servdap.legi.grenoble-inp.fr/meom/DFS5.2/.

**Author contributions**

MHR and JJ designed the research study. JJ and CCT performed the numerical simulation with inputs from OA. MHR
conducted the analysis with help from JJ. MHR wrote the manuscript with contributions from all coauthors.

**Competing interests**

The authors declare that they have no conflict of interest.

**Acknowledgments.**

We thank the IRD IMAGO team, Pierre Rousselot (ADCP), François Baurand (nutrients), and Sandrine Hillion
(chlorophyll), for collecting, validating, and making available the French PIRATA cruise measurements, along with Jacques Grelet, Fabrice Roubaud, and other engineers and technicians of the PIRATA program for maintaining the ocean-atmosphere



interaction buoys and ADCP moorings. We acknowledge the GlobColour and TropFlux projects for sharing the freely available data we use. GlobColour data has been developed, validated, and distributed by ACRI-ST, France. The TropFlux data is produced under a collaboration between Laboratoire d'Océanographie: Expérimentation et Approches Numériques
(LOCEAN) from Institut Pierre Simon Laplace (IPSL, Paris, France) and National Institute of Oceanography/CSIR (NIO, Goa, India), and supported by Institut de Recherche pour le Développement (IRD, France). TropFlux relies on data provided by the ECMWF Re-Analysis interim (ERA-I) and ISCCP projects. Supercomputing facilities were provided by GENCI project GEN7298. We acknowledge C. Ethé from the NEMO team for his help in setting up the configuration. This paper is dedicated to the memory of Christine Carine Tchamabi.

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
