# Peer review of "Physical drivers of the nitrate seasonal variability in the Atlantic cold tongue"

_Biogeosciences, 2019_

## Referee Comment (RC1) · Anonymous Referee #1 · 15 Nov 2019

This paper examines the physical processes that influence the nitrate seasonal cycle in the equatorial Atlantic cold tongue region using observations (PIRATA/EGEE cruise and PIRATA mooring data) and a numerical model (NEMO + PISCES run for the tropical Atlantic). The biogeochemical results are very interesting to this physical oceanographer, and the paper reads very nicely and is well organized. In particular, I found the results about the role of vertical processes in controlling the seasonal evolution and spatial distribution of nitrate and the interplay between the low frequency advection and advection due to tropical instability waves and eddies most fascinating. The results presented here are important and with some minor revisions this paper will be suitable for publication.

Abstract: The abstract text about the different role of horizontal advection (extends and

[Figure]

shapes the bloom off equator, brings nitrate low water below mixed layer, EUC brings low-nitrate water but still rich enough) on lines 19-23 seems a little contradictory and some wordsmithing could be applied to make clear the competing roles of zonal and horizontal advection. In contrast, the description of vertical advection and diffusion were clear.

Figure 1. Possibly enlarge Figure 1 and/or make SST contours darker/thicker so that they are easier to read.

Line 49: Suggest "1960s, as well as satellite measurements since the 1980s" instead of "60s and satellite measurements"

Line 79 (and elsewhere): Suggest acronym "TIWs" instead of "TIW"

Line 83: It is important to note here or elsewhere in the paper that TIWs exhibit seasonal variability similar to the nutrient seasonal cycle (specifically they are present with peak variance in May-July and sometimes re-emerge and there is a secondary peak in variance in the fall). This is in response to seasonal changes in the winds and the background circulation (which is drive the low-frequency vertical advection signal) but might also be contributing to the eddy vertical advection signal. Might be good to cite a study or two that shows evidence of this reemergence of TIWs in the tropical Atlantic (Caltabiano et al. 2005 OS, Athie and Marin, 2007 JGR; Perez et al. 2019 JGR, . . .)

Line 85 (and elsewhere): wording "low productive and productive seasons" is unclear, and you could perhaps switch the order or use something like "low productivity and high productivity seasons"

Line 90: Here do you mean "equatorial upwelling" instead of "equatorial divergence" since you are specifically referring to vertical processes in the parentheses?

Line 133-134: Perhaps indicate here or in Table 1 which years you include in the "low productive"/"no upwelling" averages and the "productive"/"upwelling" averages for Figure 2a-c. Are they cruise transect composites for years which productive vs. not productive but in the same season?

Line 135: Suggest "1970s and 1980s" instead of "70s and 80s"

Line 181: Suggest "three-dimensional" instead of "three-dimension"

Line 192: The second term on the right hand side in equation (2) is just the eddy part of the advection rather than the residual (sum of three terms involving an eddy term) that you describe using in the text.

Line 196: Suggest "residual" instead of "residue"

Line 202: Please identify which term in equation 1 corresponds to the "entrainment term" either on this line, or earlier in the discussion of equation 1 terms.

Line 224: "Too elevated" reads awkwardly. Please consider rewording.

Figure 4: Panel e makes it easier to compare Z20 and ZEUC between model and observations. Could a similar line plot be used to compare depth of the nitracline and DCM between model during the "no upwelling season"? It could be a panel f and fill the white space.

Line 261: There is compensation between zonal and meridional advection. Question: Which term wins and during which time of year? Which term is most responsible for bringing nitrate low waters to the cold tongue area, presumably zonal advection?

Line 281. In December, is the compensation between meridional and zonal advection different than in July? How do they contribute to the secondary maxima?

Line 310 Suggest "as an interplay" instead of "as interplays"

Line 331: Don't you mean zonal advection (rather than horizontal advection) removes nitrate all year long? Meridional advection seems to add nitrate in the mixed layer, at least in Figure 7e. You do say this later, but the statement on line 331 is in conflict with that.

Line 322-340: Some of Figure 7 labels are shifted (e.g., Fig. 7c instead of Fig. 7d) in the text in these 2 paragraphs.

Lines 358, 360. Two sentences begin with "Its" and "It" and I'm not 100% whether "It" means the EUC or nitrate concentrations or something else.

Line 447-449: The sentence beginning with "Although..." is a bit unclear as written.

General question that pertains to the last two sections in the text: How strong or realistic are the TIWs in the model? In the real ocean, do you think the eddy contribution to advection will be basically the same as what you found in the model?
* * *

---

## Referee Comment (RC2) · Anonymous Referee #2 · 18 Nov 2019

The paper presents an analysis of the nutrient supply mechanisms in the Atlantic cold tongue (upwelling system) based on a combination of a regional biogeochemical model and observations from cruised conducted over about a decade, a mooring and satellite remote sensing (chlorophyll). After showing that observations and model results agree to a good extend the authors make use of the model (output) to disentangle the role of horizontal and vertical advection and diffusion, respectively to support the observed seasonality of chlorophyll with a strong maximum in August and September and a more moderate maximum in November-December. Vertical advection and vertical diffusion are found to be is the major source terms of nitrate to the euphotic zone in the cold tongue in summer, while meridional advection redistributes nitrate (in the ml) away from the upwelling center. The difference between the stronger summer nitrate supply

(and bloom) and the smaller November-December upwelling (and bloom) are found to be associated with differences of the vertical locations of EUC core.

The paper is very well written and the descriptions are usually very clear. The study is carefully conducted and presents a important piece of science.

I have only a few minor, technical, comments:

a) The terminus 'cold tongue' is never defined, characterized or regionally narrowed down. After using this term in title and abstract, I would have expected something like a definition in the introduction. Instead you use the 'synonym' equatorial upwelling system there and only in line 79 use that terminus again. The first implicit definition that I see is in l 125ff. Perhaps it could help a wider audience if you better introduce/integrate the two terms 'cold tongue' and 'upwelling' in the introduction already.

b) l 134, Fig. 2a,b,c: please give (in the caption) explicitly which time periods you selected for no-upwelling vs. upwelling

c) l 163ff&174: boundary conditions: can you briefly explain why you mix model output and observations concerning the boundary conditions; (I am not familiar are GLORY S2V2; is this only physics?)

d) line 182, equ. 1 gives the explicit terms. Do the explicit terms at any time sum o fht dNO3/dt? What about implicit terms, i.e. transports associated with, e.g., the choosen advection scheme

e) line 201ff: a few more sentences describing the method could help here (otherwise we may need to read Vialard & Delecluse first in order to understand your quantification of entrainment; at least I did not get from that paragraph what you did)

f) caption Fig. 3; for (a) it is not clear whether surface data are shown; also the language and typographic of the phrase 'averaged in 1.5dg S-0.5deg N' can be improved

g) l 234: English: 'vertical structures are too shallow'

h) l 243: should it read: processes driving the seasonal variations of nitrate in the mixed layer are presented' ?

i) Fig. 5, caption. Please add which convention you used: 'positive eastward (c), northward (d) and upward (e,f)', I guess?

j) l 304: 'thermocline core' is not defined (from Fig. 7a I assume that you take the 20degC isothermal for the thermocline core, please say so explicitly

k) discussion: discussing the literature you discuss the role of TIW and Kelvin waves; can you make this more explicit from/for your model output ? (but I am clearly not an expert here!) from the last sentence of the conclusions though, I take that you may do so in a follow up study

Very nice, thanks!

---

## Author Comment (AC1) · 13 Dec 2019

**Response to referee #1**

*This paper examines the physical processes that influence the nitrate seasonal cycle in the equatorial Atlantic cold tongue region using observations (PIRATA/EGEE cruise and PIRATA mooring data) and a numerical model (NEMO + PISCES run for the tropical Atlantic). The biogeochemical results are very interesting to this physical oceanographer, and the paper reads very nicely and is well organized. In particular, I found the results about the role of vertical processes in controlling the seasonal evolution and spatial distribution of nitrate and the interplay between the low frequency advection and advection due to tropical instability waves and eddies most fascinating. The results presented here are important and with some minor revisions this paper will be suitable for publication.*

We thank the referee for his/her careful reading and comments. Below are our responses to the different points. Changes in the manuscript are also indicated.

*Abstract: The abstract text about the different role of horizontal advection (extends and shapes the bloom off equator, brings nitrate low water below mixed layer, EUC brings low-nitrate water but still rich enough) on lines 19-23 seems a little contradictory and some wordsmithing could be applied to make clear the competing roles of zonal and horizontal advection. In contrast, the description of vertical advection and diffusion were clear.*

We changed lines 19-23 to "Below the mixed layer, observations and model show that the Equatorial Undercurrent brings low-nitrate water (relatively to off-equatorial surrounding waters) but still rich enough to enhance the cold tongue productivity. Our results also give insights on the influence of intraseasonal processes in these exchanges. The submonthly meridional advection significantly contributes to the nitrate decrease below the mixed layer."

*Figure 1. Possibly enlarge Figure 1 and/or make SST contours darker/thicker so that they are easier to read.*

We thickened the SST contours.

*Line 49: Suggest "1960s, as well as satellite measurements since the 1980s" instead of "60s and satellite measurements"*

Done.

*Line 79 (and elsewhere): Suggest acronym "TIWs" instead of "TIW"*

Done.

*Line 83: It is important to note here or elsewhere in the paper that TIWs exhibit seasonal variability similar to the nutrient seasonal cycle (specifically they are present with peak variance in May-July and sometimes re-emerge and there is a secondary peak in variance in the fall). This is in response to seasonal changes in the winds and the background circulation (which is drive the low-frequency vertical advection signal) but might also be contributing to the eddy vertical advection signal. Might be good to cite a study or two that shows evidence of this reemergence of TIWs in the tropical Atlantic (Caltabiano et al. 2005 OS, Athie and Marin, 2007 JGR; Perez et al. 2019 JGR, ...)*

The semiannual cycle of activity of TIWs should be mentioned. We added this information in the first paragraph of the discussion on intraseasonal processes: "They are active in boreal summer, decrease in fall, emerge again at the end of the year with lesser intensity than in summer, and disappear in spring (Jochum et al., 2004; Catalbiano et al., 2005; Perez et al, 2019)."

*Line 85 (and elsewhere): wording "low productive and productive seasons" is unclear, and you could perhaps switch the order or use something like "low productivity and high productivity seasons"*

We used the referee's suggestion "low productivity and high productivity seasons".

*Line 90: Here do you mean "equatorial upwelling" instead of "equatorial divergence" since you are specifically referring to vertical processes in the parentheses?*

Yes. We changed to "equatorial upwelling".

*Line 133-134: Perhaps indicate here or in Table 1 which years you include in the "low productive"/"no upwelling" averages and the "productive"/"upwelling" averages for Figure 2a-c. Are they cruise transect composites for years which productive vs. not productive but in the same season?*

We clarify in table 1 which cruises are used for each period. We reworded lines 133-134 as "Vertical sections of nitrate, chlorophyll, and zonal current along 10° W measured during the PIRATA cruises and averaged separately in no-upwelling/low productivity and upwelling/high productivity seasons (table 1) are shown in Fig. 2a-c."

*Line 135: Suggest "1970s and 1980s" instead of "70s and 80s"*

Done.

*Line 181: Suggest "three-dimensional" instead of "three-dimension"*

Done.

*Line 192: The second term on the right hand side in equation (2) is just the eddy part of the advection rather than the residual (sum of three terms involving an eddy term) that you describe using in the text.*

Now, we clearly state that it is the eddy advection. We changed lines 194-197 by "The left hand side term is the monthly average of zonal advection. On the right hand side, the first term is the monthly zonal advection calculated from monthly averages of zonal current ($\bar{u}$) and nitrate concentrations ($\overline{NO3}$). The second term is the eddy advection term. It includes all the submonthly advection contributions which, in this region, may include influences of inertia-gravity waves, mixed Rossby-gravity waves, Kelvin waves, and eddies or tropical instability waves (e.g. Athié et al. 2009; Jouanno et al. 2013). It is calculated as the residual between…"

*Line 196: Suggest "residual" instead of "residue"*

Done.

*Line 202: Please identify which term in equation 1 corresponds to the "entrainment term" either on this line, or earlier in the discussion of equation 1 terms.*

Now, we specify more clearly what is the entrainment term: "We use the method described in Vialard and Delecluse (1998) to investigate nitrate budgets in the mixed layer and in the euphotic layer. An entrainment term appears when integrating Eq. (1) over a time-varying layer:

$$\frac{\partial \langle NO_3 \rangle}{\partial t} = -\langle u \frac{\partial NO_3}{\partial x} \rangle - \langle v \frac{\partial NO_3}{\partial y} \rangle - \langle w \frac{\partial NO_3}{\partial z} \rangle$$

$$+ \langle D_l(NO_3) \rangle + \frac{1}{h} \left( K_z \frac{\partial NO_3}{\partial z} \right)_{z=-h} + \langle SMS \rangle$$

$$- \frac{1}{h} \frac{\partial h}{\partial t} \left( \langle NO_3 \rangle - NO_{3\,z=-h} \right) \tag{3}$$

where brackets indicate the vertical average over the layer depth h. The last term arises from time-variations of the integration depth h. This term is often referred to entrainment at the base of the layer (e.g. Vialard and Delecluse, 1998) and computed as a residual of the other terms of Eq. (3). Here we verified that this term is small and we chose not to show it. The mixed layer depth is…"

The following figure shows that the contribution of entrainment is several order of magnitude smaller than the contribution of other trends: nitrate change rate (black), zonal advection (red), meridional advection (green), vertical advection (dark blue), vertical diffusion (light blue), SMS (purple) in the

mixed layer (a) and in the euphotic layer (b). The black dashed line is 1000 × entrainment, illustrating that entrainment is 3 order of magnitude smaller than the other trends.

[Figure]

*Line 224: "Too elevated" reads awkwardly. Please consider rewording.*

We changed to "too high simulated chlorophyll".

*Figure 4: Panel e makes it easier to compare Z20 and ZEUC between model and observations. Could a similar line plot be used to compare depth of the nitracline and DCM between model during the "no upwelling season"? It could be a panel f and fill the white space.*

Studying oligotrophic conditions is not the goal of this article and we prefer not to include such a plot. We specify now (lines 222-223) "In the equatorial zone, the position of the simulated DCM in the upper nitracline is in agreement with observations while its magnitude is more elevated by about 0.1 mg m$^{-3}$ (Fig. 4b)."

*Line 261: There is compensation between zonal and meridional advection. Question: Which term wins and during which time of year? Which term is most responsible for bringing nitrate low waters to the cold tongue area, presumably zonal advection?*

Thank you for this remark. It points out that the latitude range used in figure 5 of the submitted manuscript is wrong. When the average is applied with the correct latitude range, it is clear that variations of zonal advection drive variations of horizontal advection and that it brings nitrate low water during most of the year. The new figure 5 is the following one:

[Figure]

We changed the text to "Variations of zonal advection drive variations of horizontal advection (Fig. 5c, d) that acts to bring some low-nitrate water to the cold tongue area during most of the year. The main peak occurs in July-August and a secondary peak in December. Horizontal advection is close to zero in February-May."

*Line 281. In December, is the compensation between meridional and zonal advection different than in July? How do they contribute to the secondary maxima?*

This remark is linked to the preceding one. The reduction of vertical processes is mainly responsible for the reduction of nitrate supply in December. So, the lesser decrease in December is mentioned in line 261 instead of adding details in this paragraph.

*Line 310 Suggest "as an interplay" instead of "as interplays"*

Done.

*Line 331: Don't you mean zonal advection (rather than horizontal advection) removes nitrate all year long? Meridional advection seems to add nitrate in the mixed layer, at least in Figure 7e. You do say this later, but the statement on line 331 is in conflict with that.*

Yes, this is an oversimplification. We reordered this paragraph in order to better separate processes in the mixed layer and below. It is now written as:

"Below the mixed layer, horizontal advection (Fig. 7d, e) removes nitrate all year long. It drives the strong nitrate loss in August-September and the lesser one in December-January (Fig. 7c) when the contributions of both zonal (Fig. 7d) and meridional (Fig. 7e) advections are the largest. The contribution of the low frequency zonal advection (Fig. 8a) compares to that of the eddy advection (Fig. 8d) while the eddy signal (Fig. 8e) controls the meridional advection. Negative low frequency zonal and meridional advections indicate the transport of low-nitrate water from the west by the EUC and from the north by the low frequency southward component of the subsurface current (Perez et al., 2014). In the mixed layer, zonal advection acts to decrease the nitrate concentration and meridional advection is a weak source of nitrate. The low frequency advection of nitrate poor water from the east is the largest where the zonal nitrate gradient is the strongest. The low frequency meridional advection (Fig. 8b) reveals the influence of the equatorial cell: the northward transport of nitrate rich upwelled water dominates the meridional advection in the mixed layer on average in the 20° W-5° W, 1.5° S-0.5° N region."

*Line 322-340: Some of Figure 7 labels are shifted (e.g., Fig. 7c instead of Fig. 7d) in the text in these 2 paragraphs.*

We corrected labels 7b, 7c, 7d (now 7c, 7d, 7e).

*Lines 358, 360. Two sentences begin with "Its" and "It" and I'm not 100% whether "It" means the EUC or nitrate concentrations or something else.*

We changed to "**The semiannual cycle** of the nitracline depth follows the basin wide adjustment of the thermocline to the wind forcing via interactions between wind forced Kelvin waves and boundary reflected Rossby waves (Merle, 1980; Ding et al., 2009). **This adjustment** conditions the depth of the thermocline and associated nitracline that varies from 60 m in spring to about 20 m in July-August while the upwelling core remains in the upper part of the EUC, at 20-30 m, all year long (Fig. 4e)."

*Line 447-449: The sentence beginning with "Although..." is a bit unclear as written.*

We changed this sentence to "This simulation was initially designed to study the large scale processes and it does not allow concluding about the role of the different intraseasonal processes. However, our

results strongly suggest that large scale processes cannot totally explain the seasonal evolution of the nitrate budget and that the role of intraseasonal processes should be clarified."

*General question that pertains to the last two sections in the text: How strong or realistic are the TIWs in the model? In the real ocean, do you think the eddy contribution to advection will be basically the same as what you found in the model?*

We did not perform a specific validation of the TIW field in this study. Nevertheless, our experience from previous studies with this model configuration (NEMO, ¼ and 75 vertical levels) is that it reproduces the level of energy of the TIWs and their equatorial signature in terms of sea surface temperature (e.g. Athié et al. 2009; Jouanno et al. 2013). This suggests that the eddy advection contribution to the nitrate budget is well resolved. Nevertheless, this cannot be fully demonstrated from an observational basis since the only available nitrate data in the cold tongue area are from the PIRATA cruises which do not provide high-frequency information on the nutrient distribution.

We included this information in the last paragraph of the discussion. Now it reads as "This simulation was initially designed to study the large scale processes and it does not allow concluding about the role of the different intraseasonal processes. However, our results strongly suggest that large scale processes cannot totally explain the seasonal evolution of the nitrate budget. Previous studies (e.g. Athié et al.; 2009; Jouanno et al., 2013) show that this model reproduces the level of energy of the TIWs and their equatorial signature in terms of sea surface temperature. It suggests that their contribution to the nitrate budget is well resolved, but this cannot be fully demonstrated from an observational basis since the only available nitrate data in the cold tongue area are from the PIRATA cruises which do not provide high-frequency information on the nutrient distribution. A dedicated study allowing better separating the large scale and eddying signals is needed in order to identify the nature of intraseasonal processes at work and their impact on the seasonal nitrate budget in the Atlantic cold tongue area."

---

## Author Comment (AC2) · 13 Dec 2019

**Response to referee #2**

*The paper presents an analysis of the nutrient supply mechanisms in the Atlantic cold tongue (upwelling system) based on a combination of a regional biogeochemical model and observations from cruised conducted over about a decade, a mooring and satellite remote sensing (chlorophyll). After showing that observations and model results agree to a good extend the authors make use of the model (output) to disentangle the role of horizontal and vertical advection and diffusion, respectively to support the observed seasonality of chlorophyll with a strong maximum in August and September and a more moderate maximum in November-December. Vertical advection and vertical diffusion are found to be is the major source terms of nitrate to the euphotic zone in the cold tongue in summer, while meridional advection redistributes nitrate (in the ml) away from the upwelling center. The difference between the stronger summer nitrate supply (and bloom) and the smaller November-December upwelling (and bloom) are found to be associated with differences of the vertical locations of EUC core.*

*The paper is very well written and the descriptions are usually very clear. The study is carefully conducted and presents a important piece of science.*

We thank the referee for his/her remarks. We hope that our responses will clarify the different points. Changes in the manuscript are also indicated.

*I have only a few minor, technical, comments:*

a)      *The terminus 'cold tongue' is never defined, characterized or regionally narrowed down. After using this term in title and abstract, I would have expected something like a definition in the introduction. Instead you use the 'synonym' equatorial upwelling system there and only in line 79 use that terminus again. The first implicit definition that I see is in l 125ff. Perhaps it could help a wider audience if you better introduce/integrate the two terms 'cold tongue' and 'upwelling' in the introduction already.*

We agree that we should define the cold tongue at the beginning of the text. We added "Variations of the equatorial upwelling in the Atlantic Ocean are essentially seasonal. The so-called cold tongue spreads east of about 20° W to the African coast and is centered slightly south of the equator (Carton and Zhou, 1997; Caniaux et al., 2011). The maximum cooling is reached in July-August and a secondary cooling occurs in November-December (Okumura and Xie, 2006). The cold tongue is a region …" at the beginning of the introduction.

b)      *l 134, Fig. 2a,b,c: please give (in the caption) explicitly which time periods you selected for no-upwelling vs. upwelling*

Now, we explicitly indicate in table 1 which cruises are used for each period. We also reworded lines 133-134 as "Vertical sections of nitrate, chlorophyll, and zonal current along 10° W measured during the PIRATA cruises and averaged separately in no-upwelling/low productivity and upwelling/high productivity seasons (table 1) are shown in Fig. 2a-c."

c)      *l 163ff&174: boundary conditions: can you briefly explain why you mix model output and observations concerning the boundary conditions; (I am not familiar are GLORYS2V2; is this only physics?)*

Yes GLORYS2V4 is only for physics. For lateral biogeochemical forcing, we mix model and observations simply because all the variables required by the model are not available as global observational climatology. We changed the text to "Interannual atmospheric fluxes of momentum, heat, and freshwater are derived from the DFS5.2 product (Dussin et al., 2016) using bulk formulae from Large and Yeager (2009). Temperature, salinity, current, and sea level from the MERCATOR global reanalysis GLORYS2V4 (Storto et al., 2018) are used to force the model at the lateral boundaries."

d)      *line 182, equ. 1 gives the explicit terms. Do the explicit terms at any time sum o fhtdNO3/dt? What about implicit terms, i.e. transports associated with, e.g., the choosen advection scheme*

* Yes, we verified that the terms on the right hand side sum to $\partial NO3/\partial t$. We compared the variations of the monthly nitrate concentrations (black) and the nitrate time series reconstructed by integrating the right hand side over time (red) from 1995 to 2015. The following plot is an example at the surface at 15°W, 1°S. The 2 time series are superimposed and similar results are found in the thermocline.

[Figure]

* There is no implicit term for advection since we use a second order scheme without implicit numerical diffusion.

e)      *line 201ff: a few more sentences describing the method could help here (otherwise we may need to read Vialard & Delecluse first in order to understand your quantification of entrainment; at least I did not get from that paragraph what you did)*

We add some details about the calculation of the entrainment. This part is now written as "We use the method described in Vialard and Delecluse (1998) to investigate nitrate budgets in the mixed layer and in the euphotic layer. An entrainment term appears when integrating Eq. (1) over a time-varying layer:

$$\frac{\partial \langle NO_3 \rangle}{\partial t} = - \langle u \frac{\partial NO_3}{\partial x} \rangle - \langle v \frac{\partial NO_3}{\partial y} \rangle - \langle w \frac{\partial NO_3}{\partial z} \rangle$$

$$+ \langle D_l(NO_3) \rangle + \frac{1}{h}\left( K_z \frac{\partial NO_3}{\partial z} \right)_{z=-h} + \langle SMS \rangle$$

$$- \frac{1}{h}\frac{\partial h}{\partial t}\left( \langle NO_3 \rangle - NO_{3\,z=-h} \right) \tag{3}$$

where brackets indicate the vertical average over the layer depth h. The last term arises from time-variations of the integration depth h. This term is often referred to entrainment at the base of the layer (e.g. Vialard and Delecluse, 1998) and computed as a residual of the other terms of Eq. (3). Here we verified that this term is small and we chose not to show it. The mixed layer depth is…"

The following figure shows that the contribution of entrainment is several order of magnitude smaller than the contribution of other trends: nitrate change rate (black), zonal advection (red), meridional advection (green), vertical advection (dark blue), vertical diffusion (light blue), SMS (purple) in the mixed layer (a) and in the euphotic layer (b). The black dashed line is 1000 × entrainment, illustrating that entrainment is 3 order of magnitude smaller than the other trends.

[Figure]

f) *caption Fig. 3; for (a) it is not clear whether surface data are shown; also the language and typographic of the phrase 'averaged in 1.5dg S-0.5deg N' can be improved*

We changed this caption to "Figure 3: Seasonal cycle averaged in the upwelling region (a) and mean distribution in July-August (b) of simulated surface chlorophyll (mg m$^{-3}$). Data were averaged between 1.5° S-0.5° N in (a). The climatology is calculated over 1998-2015."

g) *l 234: English: 'vertical structures are too shallow'*

We changed to "vertical structures are shallower than observed"

h) *l 243: should it read: processes driving the seasonal variations of nitrate in the mixed layer are presented' ?*

We rephrased this sentence as "Understanding variations of the surface productivity requires identifying processes below the mixed layer. So in this section, processes driving the seasonal variations of nitrate are presented in the mixed layer, but also down to the base of the euphotic layer."

i) *Fig. 5, caption. Please add which convention you used: 'positive eastward (c), northward (d) and upward (e,f)', I guess?*

A nitrate tendency leads to nitrate concentration changes. Units are [concentration/time]. So, positive tendency means nitrate increase and negative tendency means nitrate decrease. It is not a nitrate transport.

j) *l 304: 'thermocline core' is not defined (from Fig. 7a I assume that you take the 20degC isothermal for the thermocline core, please say so explicitly*

It is defined in the preceding paragraph (line 193).

k) *discussion: discussing the literature you discuss the role of TIW and Kelvin waves; can you make this more explicit from/for your model output ? (but I am clearly not an expert here!) from the last sentence of the conclusions though, I take that you may do so in a follow up study*

The monthly frequency of outputs is too low to resolve the TIW and Kelvin waves (and higher frequency intraseasonal processes) satisfactorily. Higher frequency outputs and calculation of tendencies are required to better study impact of intraseasonal processes on nitrate distribution. We hope that changes in the last paragraph of the discussion make things clearer: "This simulation was initially designed to study the large scale processes and it does not allow concluding about the role of the different

intraseasonal processes. However, our results strongly suggest that large scale processes cannot totally explain the seasonal evolution of the nitrate budget. Previous studies (e.g. Jouanno et al., 2013; Athie et al., 2009) show that this model reproduces the level of energy of the TIWs and their equatorial signature in terms of sea surface temperature. It suggests that their contribution to the nitrate budget is well resolved, but this cannot be fully demonstrated from an observational basis since the only available nitrate data in the cold tongue area are from the PIRATA cruises which do not provide high-frequency information on the nutrient distribution. A dedicated study allowing better separating the large scale and eddying signals is needed in order to identify the nature of intraseasonal processes at work and their impact on the seasonal nitrate budget in the Atlantic cold tongue area."

*Very nice, thanks!*